# Deep learning radiomics can predict axillary lymph node status in early-stage breast cancer

Xueyi Zheng [1,5], Zhao Yao[2,5], Yini Huang [1,5], Yanyan Yu[3,5], Yun Wang [1], Yubo Liu [1], Rushuang Mao [1], Fei Li [1], Yang Xiao[3], Yuanyuan Wang[2,4], Yixin Hu[1], Jinhua Yu [2,4✉] & Jianhua Zhou [1✉]

Accurate identification of axillary lymph node (ALN) involvement in patients with early-stage breast cancer is important for determining appropriate axillary treatment options and therefore avoiding unnecessary axillary surgery and complications. Here, we report deep learning radiomics (DLR) of conventional ultrasound and shear wave elastography of breast cancer for predicting ALN status preoperatively in patients with early-stage breast cancer. Clinical parameter combined DLR yields the best diagnostic performance in predicting ALN status between disease-free axilla and any axillary metastasis with areas under the receiver operating characteristic curve (AUC) of 0.902 (95% confidence interval [CI]: 0.843, 0.961) in the test cohort. This clinical parameter combined DLR can also discriminate between low and heavy metastatic burden of axillary disease with AUC of 0.905 (95% CI: 0.814, 0.996) in the test cohort. Our study offers a noninvasive imaging biomarker to predict the metastatic extent of ALN for patients with early-stage breast cancer.

[1] Department of Ultrasound, Sun Yat-Sen University Cancer Center, State Key Laboratory of Oncology in South China, Collaborative Innovation Center for Cancer Medicine, Guangzhou, China. [2] Department of Electronic Engineering, Fudan University, Shanghai, China. [3] Paul C. Lauterbur Research Center for Biomedical Imaging, Institute of Biomedical and Health Engineering, Shenzhen Institutes of Advanced Technology, Chinese Academy of Sciences, Shenzhen, China. [4] The key laboratory of medical imaging computing and computer assisted intervention of Shanghai, Shanghai, China. [5]These authors contributed equally: Xueyi Zheng, Zhao Yao, Yini Huang, Yanyan Yu. ✉email: jhyu@fudan.edu.cn; zhoujh@sysucc.org.cn

Breast cancer is the most commonly diagnosed cancer among women worldwide and becomes the second leading cause of cancer-related death[1]. Accurate identification of axillary lymph-node (ALN) involvement in patients with breast cancer is important for prognosis and therapy decisions[2]. Sentinel lymph-node (SLN) is the first node draining the primary cancer. SLN dissection (SLND) is recommended to predict ALN status, especially for those with clinically negative nodes[3]. The American College of Surgeons Oncology Group Z0011 (ACOSOG Z0011) trial showed that among patients with clinical T1/T2 breast cancer, if there were two or fewer SLN metastases, the use of SLND alone would not lead to inferior survival compared with ALND[4,5]. Compared with ALND, SLND has fewer complications, but it is not risk-free surgery and still has some significant limitations, including increasing considerable anesthesia time and expense, and causing complications such as arm numbness or upper limb edema in 3.5–10.9% of patients[6,7]. There were studies showing that 43–65% of patients who had positive SLNs underwent unnecessary axillary surgery because of no additional non-SLN metastasis, resulting in high morbidity[8,9]. In fact, SLN biopsy could be avoided if there was reliable preoperative evaluation of ALN status because most patients who had early-stage breast cancer have disease-free axilla[10].

Ultrasound (US) has been widely used to preoperatively characterize breast lesions and determine ALN status[11]. A study showed that clinical T stage and preoperative axillary ultrasound (US) results were associated with the ALN status in patients with early-stage breast cancer[10], but the diagnostic performance of axillary US was poor to determine the ALN status with an area under the receiver operating characteristic curve (AUC) of 0.585–0.719 (ref. [12]). Several studies intended to predict the ALN status by clinicopathological data, such as tumor grade, histological tumor size, lymphovascular invasion, Ki-67 proliferation index, and hormone receptor status[13,14]. However, using clinicopathological data only is not accurate enough with an AUC of 0.66–0.74 in previous studies[15]. In addition, some data like lymphovascular invasion and histological tumor size could not be available preoperatively, but preoperative knowledge of ALN status is important for determining appropriate axillary treatment options[6].

Two-dimensional (2D) shear wave elastography (SWE), a new US technology to measure tissue stiffness, integrates B-mode image with a color-coded map which shows the distribution of shear wave velocity (SWV)[16]. This technique showed promise in distinguishing malignant and benign breast lesions[17]. Some studies suggested that stiffness of breast cancer was a predictor of ALN status. Higher SWV of breast cancer showed higher possibility of ALN metastasis[18,19]. However, the performance of 2D-SWE depends on the placement of regions of interest (ROI) and the AUC was only 0.759 for the prediction of ALN status[17,20]. Therefore, only SWE images of breast cancer might be insufficient to evaluate ALN status accurately.

Radiomics can automatically provide a large number of quantitative image features from medical images, which tends to be hard for naked eyes to recognize[21,22]. This method was first demonstrated to be useful in analyzing CT or MRI images on clinical oncology[23,24]. Recently, radiomics based on analysis of US images showed better performances than other routine methods[25]. However, analyzing US images by radiomics has some limitations including object segmentation and extraction of hard-coded features[22]. Deep learning radiomics, a newly developed method, can provide quantitative and high-throughput features from medical images by supervised learning[21,22]. A recent study demonstrated that this DLR was useful in analyzing SWE images and showed excellent performance in predicting the stages of liver fibrosis[22]. When applied to analyze medical images, DLR usually confronts small-sample learning problems. Clinical parameter combined DLR, which integrates clinical information with network characteristics, can help provide complementary information for image features and collaboratively use clinical information and US images features to build model, thus improving model performance[26]. Our hypothesis is that clinical parameter combined DLR might be able to extract more valuable information from images of breast conventional US and SWE and thus provide better prediction and stratification of ALN status according to the cuts off for axillary surgery of ACOSOG Z0011 trial.

Hence, the purpose of this study is to evaluate the diagnostic performance of clinical parameter combined DLR on conventional US images and SWE images of breast cancer in predicting the extent of ALN involvement in patients with early-stage breast cancer. Our results reveal that clinical parameter combined DLR yields the best diagnostic performance in predicting ALN status between disease-free axilla (N0) and any axillary metastasis (N$_+$(≥1)) with an AUC of 0.902 in the test cohort, which was significantly higher than that of axillary US ($P < 0.001$, Hanley & McNeil), classification by clinicopathological data ($P = 0.002$, Delong et al.) and DLR on images only ($P = 0.004$, Delong et al.). Clinical parameter combined DLR on breast conventional US and SWE images provides a noninvasive imaging biomarker for predicting the extent of ALN involvement preoperatively and have the potential to determine appropriate axillary treatment options for patients with early-stage breast cancer.

## Results

**Baseline characters**. Between January 2016 and April 2019, a total of 1342 women with 1342 breast lesions was studied and finally 584 women (mean age, 50 years; range, 26–83 years) with 584 malignant breast lesions were enrolled for analysis. Figure 1 shows the patient recruitment workflow. According to the results of SLND or ALN dissection, 337 had disease-free axilla (N0), 150 had low metastatic burden of axillary disease (N$_+$(1–2)) and 97 had heavy metastatic burden of axillary disease (N$_+$(≥3)).

**Base model selection and clinical information integration**. The base model acted as a feature encoder, which had a significant impact on classification. In order to find the most suitable base model for ALN prediction tasks, the performances of ResNet50, ResNet101, Inception V3, and VGG19 in predicting ALN status between N0 and N$_+$(≥1) were compared. When ResNet50 was selected as the basic model with best performance, clinical information was further added to the diagnostic model. The model incorporating clinical information was called ResNet50 + C, where C stands for clinical information. The method of adding clinical information was to directly input clinical information into the penultimate layer of the fully connected (FC) layer of ResNet50 by increasing the number of neurons. The detailed results were summarized in Table 1. The ResNet50, which integrated the deep features and clinical information offline, was proved to be the best in terms of performance and memory usage.

**Prediction of ALN status between N0 and N$_+$(≥1)**. Adopting N0 as negative reference standard, 466 lesions were randomly assigned as training cohort and the other 118 lesions as independent test cohort. The detailed characteristics including patient age, US size, Breast Imaging-Reporting and Data System (BI-RADS) category, tumor type, estrogen receptor (ER) status, progesterone receptor (PR) status, human epidermal growth factor receptor 2 (HER-2), Ki-67 proliferation index were demonstrated in Table 2. There was no significant difference between the detailed characteristics of the two cohorts (all

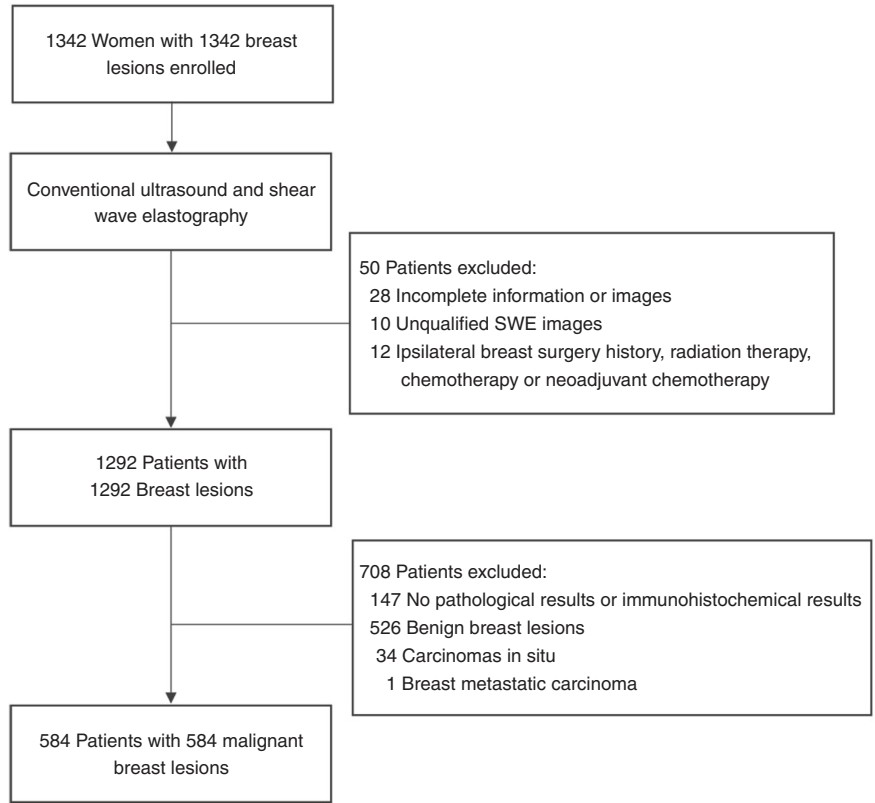

**Fig. 1 Patient recruitment workflow.** In total, 584 out of 1342 patients were included according to the selection criteria. The included patients were examined by conventional US and SWE, and had complete clinical information needed for the study.

**Table 1 The performance comparison of different models.**

| | | AUC | ACC (%) | SENS (%) | SPEC (%) | PPV (%) | NPV (%) | PARAM (Mb) |
|---|---|---|---|---|---|---|---|---|
| Resnet50 | T | 0.936 [0.910, 0.962] | 85.7 [82.0, 89.4] | 79.1 [71.6, 85.3] | 93.6 [89.2, 96.5] | 90.0 [83.5, 94.6] | 85.9 [80.6, 90.2] | 98 |
| | V | 0.904 [0.847, 0.961] | 81.4 [73.9, 88.2] | 74.0 [59.7, 85.5] | 88.2 [78.1, 94.8] | 82.2 [67.9, 92.0] | 82.2 [71.5, 90.2] | |
| | I-T | 0.902 [0.843, 0.961] | 81.0 [73.4, 87.7] | 81.6 [68.0, 91.2] | 83.6 [72.5, 91.5] | 78.4 [64.7, 88.7] | 86.2 [75.2, 93.5] | |
| Resnet50+C | T | 0.945 [0.922, 0.969] | 87.5 [84.0, 90.9] | 85.8 [80.1, 90.3] | 90.1 [85.1, 93.8] | 89.4 [84.1, 93.4] | 86.7 [81.3, 91.0] | 98 |
| | V | 0.864 [0.796, 0.933] | 75.9 [68.1, 83.6] | 73.5 [58.9, 85.1] | 88.1 [77.8, 94.7] | 81.8 [67.3, 91.8] | 81.9 [71.0, 90.1] | |
| | I-T | 0.842 [0.767, 0.916] | 74.6 [66.7, 82.4] | 70.0 [55.4, 82.1] | 76.5 [64.6, 85.9] | 68.6 [54.1, 80.9] | 77.6 [65.8, 86.9] | |
| Resnet101 | T | 0.901 [0.869, 0.934] | 81.9 [77.7, 85.8] | 72.1 [64.1, 79.2] | 92.5 [88.0, 95.8] | 87.6 [80.3, 92.9] | 81.9 [76.3, 86.7] | 172 |
| | V | 0.847 [0.771, 0.923] | 77.1 [69.0, 84.4] | 78.0 [64.0, 88.5] | 82.4 [71.2, 90.5] | 76.5 [62.5, 87.2] | 83.6 [72.4, 91.6] | |
| | I-T | 0.836 [0.758, 0.914] | 82.2 [74.3, 88.4] | 72.0 [57.5, 83.8] | 89.7 [79.9, 95.8] | 83.7 [69.1, 93.3] | 81.3 [70.7, 89.4] | |
| Inception V3 | T | 0.875 [0.841, 0.910] | 79.7 [75.2, 83.7] | 82.2 [76.2, 87.3] | 79.7 [73.5, 85.0] | 79.8 [73.6, 85.1] | 82.1 [76.0, 87.2] | 253 |
| | V | 0.853 [0.783, 0.924] | 78.5 [71.0, 85.9] | 81.6 [68.1, 91.2] | 79.1 [67.4, 88.1] | 74.1 [60.3, 85.0] | 85.5 [74.1, 93.2] | |
| | I-T | 0.796 [0.713, 0.878] | 73.7 [64.9, 80.9] | 76.0 [61.8, 86.9] | 73.5 [61.4, 83.5] | 67.9 [53.9, 79.8] | 80.6 [68.6, 89.6] | |
| VGG19 | T | 0.792 [0.744, 0.840] | 74.3 [69.7, 78.9] | 62.2 [53.8, 70.0] | 80.7 [74.6, 85.9] | 70.2 [61.6, 77.9] | 74.4 [68.1, 80.1] | 636 |
| | V | 0.759 [0.667, 0.851] | 73.3 [65.2, 81.3] | 67.4 [52.5, 80.1] | 74.6 [62.5, 84.5] | 66.0 [51.1, 78.9] | 75.8 [63.6, 85.5] | |
| | I-T | 0.750 [0.656, 0.844] | 69.5 [61.2, 77.8] | 66.0 [51.2, 78.8] | 79.4 [67.9, 88.3] | 70.2 [55.1, 82.7] | 76.1 [64.4, 85.5] | |

95% confidence intervals are included in brackets.
*AUC* area under the receiver operating characteristic curve, *ACC* accuracy, *SENS* sensitivity, *SPEC* specificity, *PPV* positive predict value, *NPV* negative predict value, *PARAM* model parameters amount, *T* training cohort (n = 350), *V* validation cohort (n = 116), *I–T* independent test cohort (n = 118).

$P > 0.05$, *t*-test or Mann-Whitney U test). Based on axillary US findings evaluated by an experienced radiologist, axillary US findings had an AUC of 0.735, accuracy of 0.635, sensitivity of 0.721 and specificity of 0.573. The Kappa values for axillary US were 0.933 for inter-observer agreement and 1 for intra-observer agreement (both $P < 0.001$, Kappa test).

In the training cohort, clinical parameter combined DLR achieved the highest AUC of 0.936 while DLR based on images only and classification by clinicopathologic data only achieved AUCs of 0.850 and 0.771, respectively. In the independent test

cohort, AUCs dropped slightly for predicting ALN metastasis and was consistent with the performance of training cohort. The clinical parameter combined DLR still achieved the highest AUC of 0.902, which was significantly higher than the AUC of other methods including axillary US findings (AUC: 0.735, $P < 0.001$, Hanley & McNeil test), DLR based on images only (AUC:0.796, $P = 0.004$, Delong et al.) and classification by clinicopathologic data (AUC:0.727, $P = 0.002$, Delong et al.). The accuracy, sensitivity, specificity, PPV, and NPV of clinical parameter combined DLR were also universally better than other methods.

**Table 2 Patient and tumor characteristics.**

| Characteristics | All patients | Training | Test | P |
|---|---|---|---|---|
| Number | 584 | 466 (80%) | 118 (20%) | – |
| Age, mean ± SD, years | 50.27 ± 10.32 | 50.46 ± 10.36 | 49.52 ± 10.20 | 0.372 |
| US size, mean ± SD, mm | 18.90 ± 6.48 | 19.06 ± 6.57 | 18.13 ± 6.10 | 0.164 |
| ER | | | | 0.567 |
| Positive | 471 | 372 (79.8%) | 99 (83.9%) | – |
| Negative | 113 | 94 (20.2%) | 19 (16.1%) | – |
| PR | | | | 0.531 |
| Positive | 429 | 338 (72.5%) | 91 (77.1%) | – |
| Negative | 155 | 128 (27.5%) | 27 (22.9%) | – |
| HER2 | | | | 0.381 |
| Positive | 135 | 104 (22.3%) | 31 (26.3%) | – |
| Negative | 449 | 362 (77.7%) | 87 (73.7%) | – |
| Ki-67 | | | | 0.380 |
| Positive | 491 | 396 (85.0%) | 95 (80.5%) | – |
| Negative | 93 | 70 (15.0%) | 23 (19.5%) | – |
| BI-RADS category | | | | 0.158 |
| 4A category | 31 | 24 (5.1%) | 7 (5.9%) | – |
| 4B category | 164 | 128 (27.5%) | 36 (30.5%) | – |
| 4C category | 267 | 210 (45.1%) | 57 (48.3%) | – |
| 5 category | 122 | 104 (22.3%) | 18 (15.3%) | – |
| Tumor type | | | | 0.742 |
| Invasive ductal carcinoma | 516 | 412 (88.4%) | 104 (88.1%) | – |
| Invasive lobular carcinoma | 18 | 12 (2.6%) | 6 (5.1%) | – |
| Other tumor types | 50 | 42 (9.0%) | 8 (6.8%) | – |

Qualitative variables are in n (%) and quantitative variables are in mean ± SD, when appropriate. Source data are provided as a Source Data file.

The detailed statistical results were summarized in Table 3 and its corresponding ROCs were shown in Fig. 2. The prediction results of DLR based on using only US or SWE images combined with clinical parameters were poorer than DLR based on using both US and SWE images combined with clinical parameters ($P = 0.006$ and $P = 0.002$, respectively, Delong et al.) (Supplementary Note 1 and Supplementary Table 2).

**Prediction of ALN status between $N_+(1-2)$ and $N_+(\geq 3)$.** Adopting $N_+(1-2)$ as negative reference standard, this experiment assigned 197 lesions as training cohort and 50 as independent test cohort. In the training cohort, DLR based on images only and classification by clinicopathologic data achieved AUCs of 0.874 and 0.756, respectively, while clinical parameter combined DLR achieved the AUC of 0.956. In the independent test cohort, the AUC of clinical parameter combined DLR dropped slightly but still reached 0.905, which was significantly higher than the AUC of DLR based on images only (AUC: 0.777, $P = 0.04$, Delong et al.) and the AUC of classification by clinicopathologic data (AUC: 0.686, $P = 0.03$, Delong et al.). The detailed statistical results were summarized in Table 4. The corresponding ROCs depicted the comparisons (Fig. 3).

**Prediction of ALN status among N0, $N_+(1-2)$ and $N_+(\geq 3)$.** This model was extended to be compatible with three groups of tasks to predict ALN status. As described above, the clinical endpoints were categorized into three parts: N0, $N_+(1-2)$, and $N_+(\geq 3)$. The number of lesions of the three categories is 337 (N0), 150 ($N_+(1-2)$), and 97 ($N_+(\geq 3)$), respectively. The DLR model was built on breast conventional US and SWE images and was classified by axillary US findings, clinicopathologic data. The overall accuracy of differentiating the three groups was 0.805 and the confusion matrix was shown in Fig. 4. The model performed well in differentiating the N0 group while showed poorer results in the other two groups.

**Interpretability of DLR model.** For investigating the interpretability of the DLR, the network was visualized by applying the Gradient-weighted Class Activation Mapping (Grad-CAM), which could produce a coarse localization map highlighting the import regions for classification target[27]. The last convolutional layer of the last res-block was made transparent to the prediction of ALN status as shown in Fig. 5. We found that there were usually two locations valuable for predicting ALN status based on DLR model. One is the boundary of the tumor and the other is the low echo area inside the tumor. To some extent, this proved the effectiveness of the model.

## Discussion

According to ACOSOG Z0011 trial, patients who had early-stage breast cancer with less than 2 SLN metastasis had no inferior survival if they underwent SLND only rather than ALN dissection[4,5]. Based on the results of ACOSOG Z0011 trial, all patients should undergo SLND to predict ALN status whether patients have clinically positive node or not[28]. However, SLND has some limitations, including causing some complications[6], having false-negative rates ranging from 7.8–27.3%[29–31] and resulting in unnecessary axillary surgery[9]. Hence, there is an increasing need for predicting metastatic extent of ALN accurately in a noninvasive way.

In this study, we developed and validated a clinical parameter combined DLR method based on breast conventional US and SWE images for preoperative prediction of ALN status in patients with clinical T1 or T2 breast cancer. This method showed significantly better diagnostic performances in distinguishing patients with a negative axilla (N0) and patients with any axillary metastasis ($N_+(\geq 1)$) than any single method. Encouragingly, our model showed favorable discriminating ability between patients with low metastatic burden of axillary disease ($N_+(1-2)$) and patients with heavy metastatic burden of axillary disease ($N_+(\geq 3)$). With false-negative rate similar to SLND, this clinical parameter combined DLR might have the potential to serve as a noninvasively imaging biomarker to replace SLND for patients with early-stage breast cancer. The clinical parameter combined DLR showed the possibility to assist breast clinicians to make decisions for appropriate axillary treatment: no need for SLN biopsy or ALN dissection in patients with N0, SLND only for patients with $N_+(1-2)$ and ALN dissection for patients with $N_+(\geq 3)$[4].

Some studies argued that compared with SLND, axillary US combined with fine needle aspiration or core needle biopsy could be cost saving for patients with positive nodal status[11]. However, axillary US was not accurate enough to predict ALN status with an AUC of 0.585–0.719 (refs. [12,32]). Some studies demonstrated that the number and morphology of abnormal lymph nodes detected by axillary US were predictors of nodal burden[28], but the diagnostic performances were poor with AUC of 0.725–0.747 (refs. [33,34]). In our study, the overall diagnostic performance of preoperative axillary US results was low with an AUC of 0.735, which was concordant with previous studies[35,36]. Compared with axillary US results alone, the clinical parameter combined DLR method makes use of all available data including findings of axillary US, clinicopathologic data, breast conventional US and SWE images and therefore, showed significantly better diagnostic performances in predicting ALN metastasis than the routine axillary US evaluated by an experienced radiologist.

Some studies reported that some histopathological data such as tumor grade, lymphovascular invasion, histological tumor size,

**Table 3 The prediction of ALN status results (N0 v.s. N$_+$(≥1)).**

| Methods | | AUC | ACC (%) | SENS (%) | SPEC (%) | PPV (%) | NPV (%) |
|---|---|---|---|---|---|---|---|
| Axillary US | | 0.735[a] [0.694, 0.775] | 63.5 [58.6, 70.4] | 72.1 [76.1, 84.8] | 57.3 [52.3, 64.9] | 72.6 [67.9, 77.2] | 57.3 [50.1, 61.2] |
| Classification by | T | 0.771 [0.719, 0.824] | 73.6 [68.8, 78.0] | 68.0 [59.8, 75.5] | 79.6 [73.4, 84.9] | 70.9 [62.7, 78.3] | 77.3 [71.0, 82.8] |
| clinicopathologic data | V | 0.755 [0.665, 0.845] | 71.6 [63.3, 79.8] | 63.3 [48.3, 76.6] | 71.6 [59.3, 82.0] | 62.0 [47.2, 75.3] | 72.7 [60.3, 83.0] |
| | I–T | 0.727[b] [0.630, 0.825] | 70.9 [62.1, 78.6] | 62.0 [47.2, 75.3] | 69.1 [56.7, 79.8] | 59.6 [45.1, 73.0] | 71.2 [58.7, 81.7] |
| DLR on images only | T | 0.850 [0.813, 0.887] | 76.7 [72.1, 81.0] | 71.6 [64.7, 77.8] | 80.2 [74.0, 85.5] | 77.9 [71.1, 83.7] | 74.3 [68.0, 80.0] |
| | V | 0.804 [0.717, 0.891] | 72.4 [63.3, 79.8] | 69.4 [54.6, 81.7] | 79.1 [67.4, 88.1] | 70.8 [55.9, 83.0] | 77.9 [66.2, 87.1] |
| | I–T | 0.796[c] [0.708, 0.883] | 71.6 [63.0, 79.4] | 67.4 [52.5, 80.1] | 79.1 [67.4, 88.1] | 70.2 [55.1, 82.7] | 76.8 [65.1, 86.1] |
| Clinical parameter | T | 0.936 [0.910, 0.962] | 85.7 [81.7, 89.1] | 79.1 [71.6, 85.3] | 93.6 [89.2, 96.5] | 90.0 [83.5, 94.6] | 85.9 [80.6, 90.2] |
| combined DLR | V | 0.904 [0.847, 0.961] | 81.4 [73.9, 88.2] | 74.0 [59.7, 85.5] | 88.2 [78.1, 94.8] | 82.2 [67.9, 92.0] | 82.2 [71.5, 90.2] |
| | I–T | 0.902 [0.843, 0.961] | 81.0 [73.4, 87.7] | 81.6 [68.0, 91.2] | 83.6 [72.5, 91.5] | 78.4 [64.7, 88.7] | 86.2 [75.2, 93.5] |

95% confidence intervals are included in brackets. Source data are provided as a Source Data file.
*AUC* area under the receiver operating characteristic curve, *ACC* accuracy, *SENS* sensitivity, *SPEC* specificity, *PPV* positive predict value, *NPV* negative predict value, *DLR* deep learning radiomics, *T* training cohort (*n* = 350), *V* validation cohort (*n* = 116), *I–T* independent test cohort (*n* = 118).
[a] indicates *P* < 0.001, Hanley & McNeil in comparison with clinical parameter combined DLR in independent test cohort.
[b] indicates *P* = 0.002, Delong et al. in comparison with clinical parameter combined DLR in independent test cohort.
[c] indicates *P* = 0.004, Delong et al. in comparison with clinical parameter combined DLR in independent test cohort.

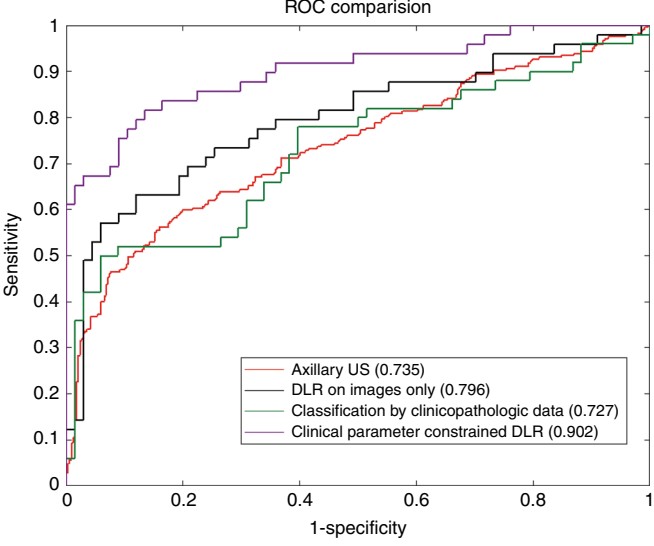

**Fig. 2 Comparison of receiver operating characteristic (ROC) curves between different models for predicting disease-free axilla (N0) and any axillary metastasis (N$_+$(≥1)).** DLR deep learning radiomics. Numbers in parentheses are areas under the receiver operating characteristic curves. Source data are provided as a Source Data file.

and hormone receptor status could be a predictor of ALN metastasis[10,37]. However, some histopathological data, such as histological tumor size and lymphovascular invasion, could only be evaluated after surgical resection, and could not guide decisions of axillary surgery preoperatively. Although tumor grade could be estimated from core biopsy samples preoperatively, low concordant rates ranging from 67% to 75% were found between core needle biopsy and surgical excision in previous studies[38,39]. Different from previous studies, this current study adopts all histopathological data available after biopsy of primary breast tumor[40], which is a standard procedure preoperatively. Therefore, some clinicopathologic data available preoperatively were kept as candidate factors in developing the predictive model, which could serve as a noninvasive predictive method to assess ALN status.

Clinical parameter combined DLR was completely established on analyzing images of breast conventional US and SWE with the DLR concept[22] and was combined with axillary US findings and clinicopathologic data. This DLR method has shown great promise in analyzing SWE images on staging liver fibrosis[22].

Radiomics method was also applied in other imaging modalities like CT or MRI images of some primary cancer like bladder, colon cancer to predict regional lymph-node metastasis, demonstrating this method was a useful way to make a prediction of lymph-node metastasis[41,42]. Compared with the previous study, our study yielded a better diagnostic performance by concentrating on the clinical parameter combined DLR method, which can complement image features with more information and make the model more robust by restraining the features extracted from images[26]. In addition, for patients with suspected breast lesions, breast and axillary US is a routine practice to characterize breast lesions and axillary lymph-node status, and have the advantages of cost effective and no ionizing radiation comparing with other imaging modalities[10].

Compared with those studies using SWE values as a single parameter to predict ALN status[12,18], our study showed a better diagnostic performance by applying DLR on breast conventional US and SWE images. Instead of measuring the stiffness of breast cancer inside the shear wave ROI based on several parameters of SWE, the whole shear wave ROI was analyzed and a large number of features were quantified automatically by DLR[22].

Some limitations have to be addressed in this study. First, this is a single-center study. Acquiring more evidence from multi-center is needed to validate this model before clinical application in the future. Second, patients with multifocal breast lesions and bilateral disease are excluded because it is difficult to determine which lesion would lead to ALN metastases and should be input in the model. Therefore, current clinical parameter combined DLR model could only be used to predict extent of ALN involvement for patients with single breast cancer. Further study is needed to build other model to predict ALN status for patients with multifocal breast lesions and bilateral disease. Third, gene markers of breast cancer like BRCA1 and BRCA2, are used to stratify patients based on the risk for disease[43]. However, radiogenomics, focusing on the relationship between genomics and imaging phenotypes, is not available currently although it is an interesting attempt.

## Conclusions

Clinical parameter combined DLR on breast conventional US and SWE images provides a noninvasive and practical way for predicting the extent of ALN involvement preoperatively and have the potential to determine appropriate axillary treatment options for patients with early-stage breast cancer. Prospective multi-center validation is expected to acquire high-level evidence for clinical use in subsequent studies.

**Table 4 The prediction of ALN status results ($N_+$(1–2) v.s. $N_+$($\geq$3)).**

| Methods | | AUC | ACC (%) | SENS (%) | SPEC (%) | PPV (%) | NPV (%) |
|---|---|---|---|---|---|---|---|
| Classification by clinicopathologic data | T | 0.756 [0.674, 0.838] | 73.9 [66.6, 80.7] | 65.5 [51.9, 77.5] | 75.6 [65.4, 84.0] | 63.3 [49.9, 75.4] | 77.3 [67.1, 85.5] |
| | V | 0.701 [0.567, 0.835] | 72.0 [58.8, 84.1] | 62.0 [36.1, 80.9] | 78.0 [61.4, 92.3] | 66.7 [40.2, 87.1] | 75.0 [56.6, 88.5] |
| | I–T | 0.686[a] [0.528, 0.844] | 71.4 [57.3, 82.7] | 68.4 [43.4, 87.4] | 73.3 [54.1, 87.7] | 61.9 [38.4, 81.9] | 78.6 [58.6, 91.9] |
| DLR on images only | T | 0.874 [0.814, 0.934] | 79.7 [72.5, 85.6] | 84.5 [72.6, 92.7] | 80.0 [70.2, 87.7] | 73.1 [60.8, 83.3] | 88.9 [79.9, 94.8] |
| | V | 0.80 [0.671, 0.929] | 73.5 [61.1, 85.8] | 84.2 [60.4, 96.6] | 73.3 [54.1, 87.7] | 66.7 [44.7, 84.4] | 88.0 [68.8, 97.5] |
| | I–T | 0.777[b] [0.644, 0.911] | 69.4 [55.0, 80.9] | 79.0 [54.4, 93.9] | 66.7 [47.2, 82.7] | 60.0 [38.7, 78.9] | 83.3 [62.1, 95.4] |
| Clinical parameter combined DLR | T | 0.956 [0.926, 0.986] | 89.2 [84.2, 94.2] | 91.4 [81.0, 97.1] | 87.8 [79.2, 93.7] | 82.8 [73.1, 91.1] | 94.0 [86.6, 98.1] |
| | V | 0.925 [0.850, 0.997] | 88.0 [78.6, 96.9] | 95.0 [75.1, 98.9] | 86.7 [69.1, 95.3] | 82.6 [60.6, 95.2] | 96.3 [81.0, 98.0] |
| | I–T | 0.905 [0.814, 0.996] | 80.0 [68.9, 91.1] | 85.0 [62.1, 96.8] | 86.7 [69.3, 96.2] | 81.0 [57.4, 94.8] | 89.7 [72.6, 97.8] |

95% confidence intervals are included in brackets. Source data are provided as a Source Data file.
*AUC* area under the receiver operating characteristic curve, *ACC* accuracy, *SENS* sensitivity, *SPEC* specificity, *PPV* positive predict value, *NPV* negative predict value, *DLR* deep learning radiomics, *T* training cohort (n = 148), *V* validation cohort (n = 49), *I–T* independent test cohort (n = 50).
[a]Indicates P = 0.03, Delong et al. in comparison with clinical parameter combined DLR in independent test cohort.
[b]Indicates P = 0.04, Delong et al. in comparison with clinical parameter combined DLR in independent test cohort.
Source data are provided as a Source Data file.

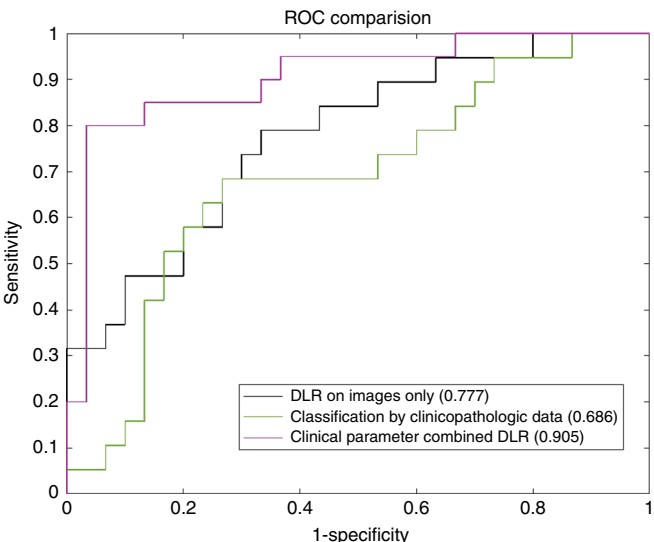

**Fig. 3 Receiver operating characteristic (ROC) curves comparison between different models for predicting low metastatic burden of axillary disease ($N_+$(1–2)) and heavy metastatic burden of axillary disease ($N_+$($\geq$3)).** DLR deep learning radiomics. Numbers in parentheses are areas under the receiver operating characteristic curves. Source data are provided as a Source Data file.

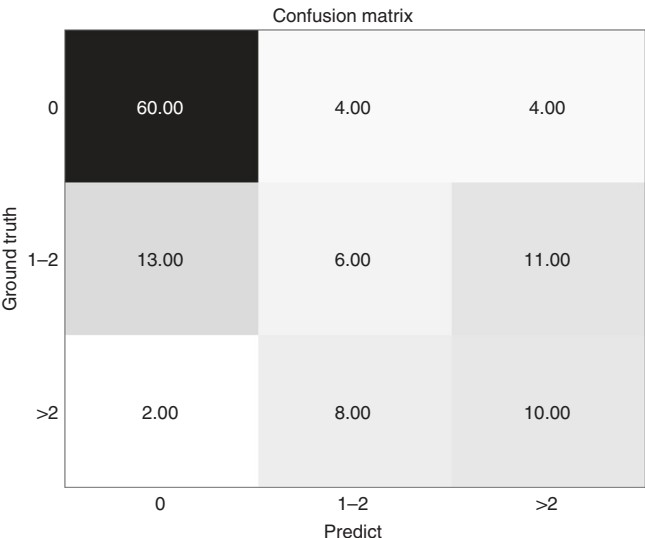

**Fig. 4 The confusion matrix of predicting metastasis among disease-free axilla (N0), low metastatic burden of axillary disease ($N_+$(1–2)) and heavy metastatic burden of axillary disease ($N_+$($\geq$3)).** Source data are provided as a Source Data file.

## Methods

**Patients.** This prospective study was approved by Institutional Review Board of Sun Yat-sen University Cancer Center. The inclusion criteria included the followings: (a) women with US-suspected breast masses; (b) availability of clinical data; (c) patients who underwent breast surgery and sentinel lymph-node biopsy or ALN dissection with curative intent. The exclusion criteria included the followings: (a) preoperative therapy (resection biopsy, neoadjuvant radiotherapy or chemotherapy); (b) patients with multifocal lesions or bilateral disease; (c) masses deeper than 3 cm in depth due to the attenuation of SWE or larger than 3.5 cm in diameter due to the limited width of the US probe; (d) unqualified 2D-SWE measurements, which means little or no shear wave signal was acquired in the ROI of SWE; (e) benign breast lesions or carcinoma in situ; (f) missing important histopathological results (immunohistochemical results or lymph-node results); (g) incomplete information or images. Verbal informed consent was obtained from all patients.

**Conventional US examinations.** One of five radiologists who had 18, 6, 3, 2, and 2 years of experience in breast ultrasound respectively performed preoperative breast and axillary US with Siemens S2000 ultrasound scanner (Siemens Healthineers, Mountain View, CA, USA) equipped with a 4–9 MHz linear array transducer. The target breast mass was measured at maximal-diameter plane to determine US size and classified by using US BI-RADS[17]. After performing whole-breast US, the same

radiologists performed axillary US routinely and recorded suspicious US features of ALN. Suspicious US features of ALN include the ratio of long axis diameter to short axis diameter<2, diffuse cortical thickening>3 mm, focal cortical bulge >3 mm, eccentric cortical thickening >3 mm, rounded hypoechoic node complete or partial effacement of the fatty hilum, nonhilar cortical blood flow on color Doppler images, complete or partial replacement of the node with an ill-defined or irregular mass and microcalcifications in the node[32]. The result of axillary US evaluated by the experienced radiologist was regarded as positive as long as at least one suspicious US finding was found. The result of axillary US was regarded as negative when no suspicious findings of ALN were found[10]. To evaluate the intra-observer agreement for axillary US, one radiologist repeated evaluating the same 30 ALNs at a time interval of 1 week. Inter-observer agreement was tested by two radiologists, evaluating the same ALNs independently in another 30 ALNs.

**SWE.** After performing conventional US, SWE was performed thrice at the maximal-diameter plane of the breast lesion. The ROI of SWE was adjusted to include subcutaneous fat layer and superficial pectoral muscle layer, with at least 5 mm of distance from the boundary of the lesion to the lateral borders[44]. With sufficient coupling material filling between probe and skin, the radiologist applied extremely slight pressure to minimize pre-compressions. When acoustic radiation force impulse was generated, patients were asked to suspend respiration for several seconds. The quality map, displayed in red-yellow-green representing low-intermediate-high quality respectively, was obtained to evaluate the quality of the SWE first. Then the velocity map of SWE was obtained. Guided by the quality map,

the velocity map of SWE with fewest artifacts and the best quality was chosen and stored as the SWE image for analysis.

**Data analysis**. Clinical and histopathologic data were obtained from the medical records. Histopathologic results of the breast cancer included tumor type, ER status, PR status, HER-2, and Ki-67 proliferation index. Clinical data included patients age, US size, tumor location, and BI-RADS category. Histopathologic

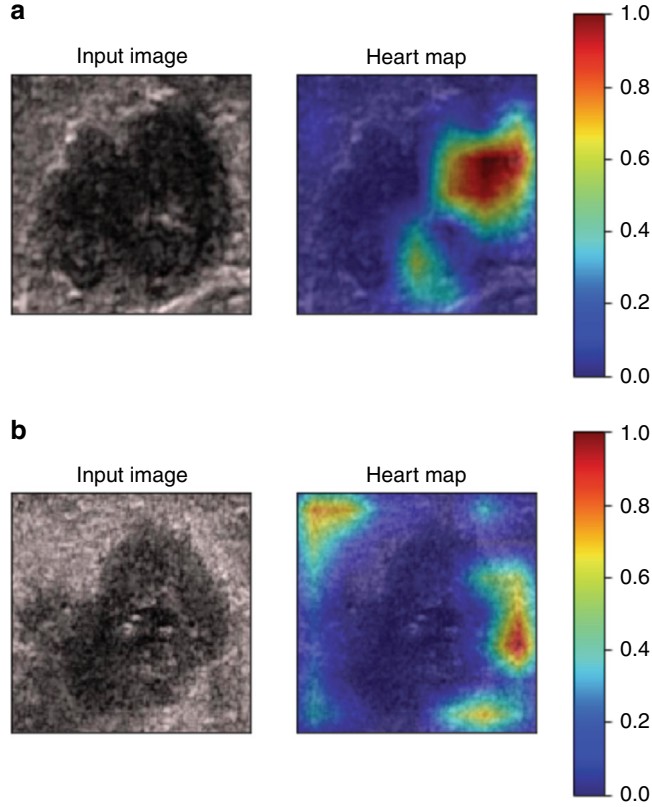

**Fig. 5 Visualization of two patient examples.** Each example shows the gray-scale US image and corresponding heart map, and the red region represents a larger weight, which can be decoded by the color bar on the right. Image a shows that the low echo area inside the tumor is valuable for predicting ALN status, while it is the tumor boundary for image b.

results of SLND and ALN dissection including the total number of resected lymph nodes and total number of positive nodes were recorded.

**Deep learning radiomics model**. The enrolled patients were randomly divided into the training cohort and independent test cohort with the ratio of 4:1 and the training cohort were then used to optimize the model parameters. We also randomly chose 25% of training images to form a validation cohort to guide the choice of hyper parameters. The whole pipeline of our model was shown in Fig. 6. Resnet was adopted as the base model which pre-trained on Imagenet[45,46]. In particular, the last 1000 nodes FC layer was replaced with our specifically designed three FC layers with Xavier initialized weights[47]. The detailed architecture of the network is shown in Supplementary Table 1.

There were two steps included in the entire process, the forward computation and the backward propagation[48]. Before that, the rectangular ROIs were cropped from raw US images according to the tumor segmentation mask, resized to 224 × 224 pixels and normalized. The pathology type was encoded to one-hot, which was the label. In the training stage, rectangular ROIs were fed into network to update model parameters by backward propagation. The outputs of the network were used as the classification results, and the cross-entropy of the outputs and the labels were calculated as the loss function. Note that to alleviate the influence of over fitting and sample imbalance, a strategy called online data augmentation was used, which meant randomly horizontal and vertical flipping the input image, randomly cropping every ROI image from four directions in the steps of 2 pixels and feeding each category image into the network with same probability. We set learning rate to 1e-4 and applied the Adam optimizer to update the model parameters with batch size 32. The maximum iteration step was set to 5000, and the learning rate decayed by 1/2 at 2000 and 4000 steps. After training, we replaced the last FC layer with an SVM as classifier and fused the clinical information and network features to collaboratively make a decision[49].

**Network feature extraction**. Contrasting with hand-crafted and engineered features designed according to the previous medical experiences, DLR learnt the high-throughput image features in a supervised manner, which could make full use of all embedded information in US images[50,51]. The convolutional layers encoded the input rectangular ROIs and adaptively learnt the semantic features and the FC layers then selected the relevant features and reduced the features dimensions. Supervised by the label of input images, the model updated parameters and finally led to the most relevant features in the FC layer. The penultimate FC layer output was used as the network features, which further proved efficient and effective. For comparison, clinicopathological information and network features were used to train SVMs directly to compare predictive performance of ALN status, respectively.

**Multi-modal multi-sources features fusion on DLR**. As described earlier, a single modal image could be encoded into network features. An additional modal image might provide more effective information. Hence the model was extended to be compatible with bimodal image inputs. A parallel model structure was designed, which contained two ResNet50 base networks and FC layers. The two ResNet50 base model shared parameters, which accepted US and SWE images as bimodal image inputs, respectively. The final convolutional layer outputs of the parallel network were concatenated and fused by the following FC layers. Network features were extracted from layers the same as one modal model. We argued that when

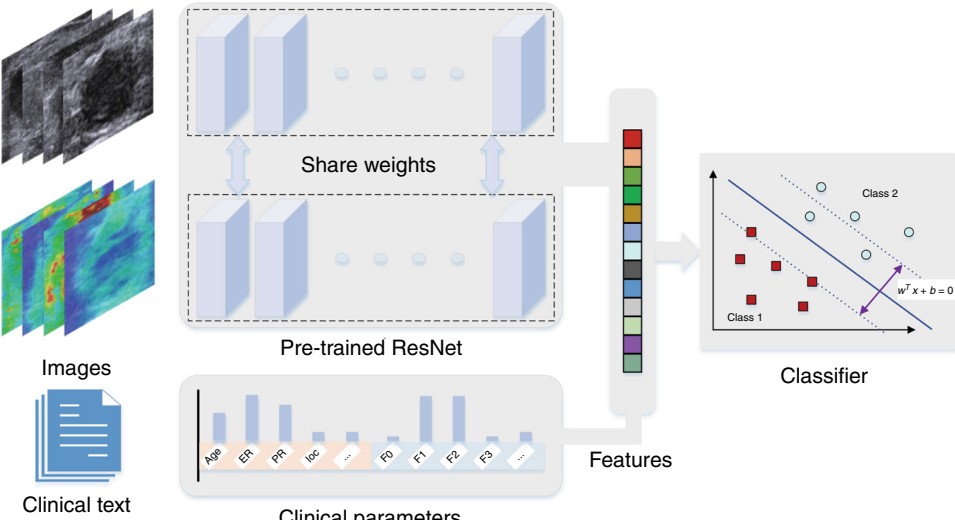

**Fig. 6 The overall pipeline of the model.** The parallel pre-trained ResNet model encodes the input images to features which be combined with clinical parameters. Then the combined features be classified by an SVM model.

deep models were applied to medical images analysis usually confronting small-sample learning problems, they should be combined with the clinical information[26]. Finally, the deep features combined with clinical features were used to collaboratively train an SVM classifier for predicting ALN status. In the control group model ResNet50+C, the neurons were added to the penultimate FC layer of the ResNet50 with the same number of clinical features. During the model training, the network received dual-modal images and as the input, and the clinical features were directly input into the penultimate FC layer instead of extracting deep features and combining them offline. To use the proposed model, a rectangle ROI which cover the tumor should be manually selected as the network input.

**Statistical analysis.** By using SLN biopsy or ALN dissection as reference standard, the extent of ALN metastasis was divided into three groups, including disease-free axilla (N0), low metastatic burden of axillary disease ($N_+(1–2)$) and heavy metastatic burden of axillary disease ($N_+(\geq 3)$). The detailed clinicopathological difference of N0 and $N_+(\geq 1)$ was compared by t-test or Mann-Whitney U test. AUC was used to estimate the performance of axillary US, DLR based on images only, classification by clinicopathological data and clinical parameter combined DLR, and was compared by using Delong et al[52]. or Hanley & McNeil[53,54]. The other measurements like accuracy, sensitivity, specificity, positive predictive value (PPV), and negative predictive value (NPV) were also used to estimate the model performance. The calculation method is shown in Supplementary methods. Kappa test was used to compared the intra-observer agreement and inter-observer agreement. All the statistics were two side and a P-value less than 0.05 was considered statistically significant. All statistical analyses were performed using MedCalc software (V.11.2; 2011 MedCalc Software bvba, Mariakerke, Belgium), Python 3.5, matlab R2015b.

**Statistics and reproducibility.** Models were verified and replicated using regular machine learning metrics on independent test cohorts. We released the software of the model for replication on new data.

**Reporting summary.** Further information on research design is available in the Nature Research Reporting Summary linked to this article.

## Data availability
Excel files containing raw data included in the main figures and tables can be found in the Source Data File in the article. All other data are available in the Article and Supplementary Information. All other data including the imaging data can be provided upon reasonable request to the corresponding author.

## Code availability
The software and code of the proposed method are available as Supplementary Software.

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

## Acknowledgements

The work was supported by National Key R&D Program of China [No. 2019YFC0118300], National Natural Science Foundation of China [No. 81971631], Major Research plan of the National Natural Science Foundation of China [No. 91959127], Shanghai Municipal Science and Technology Major Project [No. 2018SHZDZX01].

## Author contributions

J.H.Z. and J.H.Y. conceived and designed the project; Y.B.L., Y.W., Y.N.H., F.L., R.S.M., Y.X. H., and X.Y.Z. performed the research and collected the data; Z.Y., Y.Y.Y., Y.X., and Y.Y.W. analyzed the data; Z.Y., Y.Y.Y., and J.H.Y. proposed the model; X.Y.Z. Z.Y., J.H.Y., and J.H.Z wrote the paper. All authors read and approved the final version of the article.

## Competing interests

The authors declare no competing interests.
