## [Peer Review File · Nature Communications]

Reviewers' Comments:

Reviewer #1:

Remarks to the Author:

An interesting study that uses deep learning features to classify axillary lymph nodes.

Pg 3, line 40 - seems to suggest that breast cancer is the leading cause of cancer related deaths in women, however as noted in ref 1, it is lung cancer.

The term "clinically regularized DLR" should be better define- adding clinical parameters to imaging features as input to a classifier is not "regularization" in the classical sense of the term.

p3, line 42- Did you mean sentinel node biopsy? Potentially split into two sentences, one that describes what a sentinel node is and what is recommended?

pg 4, line 84 -please explain what self-regulated learning is

It is not clear why you did not just add the clinical features directly to the neural network, potentially just before the last layer . It would be good to compare the performance of such an architecture with the one proposed here.

pg 6, line 106- can you comment on this limitation? What fraction of patients would be excluded in general?

pg 6, line 114 - what was the inter-rater agreement?

Did you pre-process your images (besides cropping)? Normalize? Please provide full details of the preprocessing step.

pg 7, line 154 ..the last 1000 nodes fully connected layer was replaced with our delicate designed three fully connected layers with Xavier initialized weights" - what does delicate design refer to? Could you also comment on the reason for the choice of this network architecture? Did you try others? Please provide sufficient technical details regarding the training process.

Can you further clarify the training of the network? Specifically, what did the label consist of? The ground truth for the classification (i.e. one of 3 classes)? What were the range of sized for the ROIs?

pg 8, line 182 -this section is not clear.

Please comment on the required performance metrics that would make the sentinel node biopsy unnecessary.

There were numerous linguistic issues:

pg 3, line 47 -please define SLND

There are a number of run-on sentences that could be split up for improved clarity or text that could be rephrased to improve readability including:

pg 1, line 14-16

pg 3, line 59

pg 3 line 63-65

pg 4, line 68 This technique showed promising... -> ...showed promise...

pg 4, line 69

pg 4, line 69

pg 4, line 75-77. not sure what "quantificationally" means
pg 4, line 90 "... restrain the features extracted from images to make the model more robust." Not sure what "restrain" refers to in this context or how this process makes the model more robust.
pg 5- not sure why this is blank?
pg7, line 138 ...thought as unqualified... ->rephrase
pg7, line 156 rephrase "enrolled"
pg 8, line 183 - what is "modal" in this context?
pg 8, line 187 -"fc" - please spell out fully connected the first time you use it.
pg 9, line 198 -By using different cut-offs of positive ALNs as reference -> please expand what cut-offs refer to?
pg 11, line 243

Reviewer #2:

Remarks to the Author:

Introduction

Line 48, SLN is not controversial, it is standard care at present. Time and expense when combined with resection of the primary tumour is modest.

Line 50 "Injury" should be replaced with "complication" and the frequency of the side effects should be stated

Line 52 replace "can" with "could" and is with was

Line 64 be more specific , it is only vi status in the list you give which is unknown pre-op

Line 77 quantificationally is a clumsy word , please rephrase

Line 78 poor English suggest "when analysing oncology CT and MRI images" replace "upon" with "on" x2

Line 81 replace "have" with "has"

Methods

108 please explain what "unqualified SWE images" means

Were these consecutive lesions? If not, exactly who was recruited and why?

It is not believable that no patient had multiple lesions, in my experience 10-20% of breast cancer patients have multifocal disease on US and 2% have bilateral disease, please explain, if multifocal disease was excluded please state, if the largest lesion only was included please state. If so this needs discussion.

Were men excluded?

Were they symptomatic or mammography detected ?

How many SWE images were performed in how many planes, cancer show marked anisotropy so was any effort made to use the stiffest or softest plane? Multiple images have been shown to increase the reproducibility of the technique.

Results

Please give mean age and age range

Line 227 remove "poor" results are data not opinion

Clinical data should include only data available pre-operatively. IE core grade not final histological grade, and exclude VI status, US size not histological size.

In patients with a pre-operative diagnosis of axillary metastases (fna or core) how good was prediction of nodal burden at surgery? Especially those who had ANC.? This is a crucial clinical question.

Discussion

The ZOO11 trial had a number of well publicised limitations which should be mentioned

A few recent papers have addressed predicting nodal burden with US , these should be mentioned

Line 317 tumour grade is routinely estimated from core biopsy samples pre-operatively in most centres , this should be addressed.

How would DLR be used in a clinical setting, would an online tool be feasible? Would DLR be possible on uploaded images? How long would it take?

Table 1

HER positivity is usually seen in 10-20% of patients and ER positivity in about 80% , your results especially for HER-2 status of <1% are not credible

The percentages in the BIRADS section have been wrongly calculated , each column not line should add up to 100%

The location section should be removed

The tumour type (ie ductal , lobular etc) should be added

October 17, 2019

RE: NCOMMS-19-24501A

TITLE: Deep learning radiomics of conventional ultrasound and shear wave elastography in early-stage breast cancer: preoperative prediction of axillary lymph node status.

Dear reviewers,

Thank you for reviewing our manuscript entitled "Deep learning radiomics of conventional ultrasound and shear wave elastography in early-stage breast cancer: preoperative prediction of axillary lymph node status". As per your request, this letter will address all the raised criticisms. Modifications were made and highlighted in the revised text in response.

Reviewers' comments and answers:

Reviewer #1:

Q 1: Pg 3, line 40 - seems to suggest that breast cancer is the leading cause of cancer related deaths in women, however as noted in ref 1, it is lung cancer.

A: This had been corrected accordingly and it had been pointed out that breast cancer was the second leading cause of cancer related deaths in women. Please see page 3, line 39-40.

Q 2: The term "clinically regularized DLR" should be better define- adding clinical parameters to imaging features as input to a classifier is not "regularization" in the classical sense of the term.

A: According to your suggestion, regularization is indeed not suitable here, so we have changed 'clinically regularized DLR' into 'clinical parameter constrained DLR', which integrated clinical information and

network characteristics. Please see page 4, line 89-92.

Q 3: p3, line 42- Did you mean sentinel node biopsy? Potentially split into two sentences, one that describes what a sentinel node is and what is recommended?

A: This sentence had been split into two sentences, one describing what a sentinel node is and the other describing what is recommended. Please see page 3, line 42-44.

Q 4: pg 4, line 84 -please explain what self-regulated learning is

A: In the revised manuscript, we had changed the less-used term 'self-regulated learning' to the widely accepted term 'supervised learning'. What we originally wanted to express was to extract image features by learning rather than manual design. Please see page 4, line 85-86.

Q 5: It is not clear why you did not just add the clinical features directly to the neural network, potentially just before the last layer. It would be good to compare the performance of such an architecture with the one proposed here.

A: As a comparison, we listed the performance of model used in manuscript and other models, including the model adding the clinical features directly to the last but one fully connected layer (Resnet50_concatenate), Resnet101, Inception v3 and vgg19. All model performance is for tasks that predicting ALN metastasis between disease-free axilla (N0) and any axillary metastasis (N+(≥1)). Evaluation indicators include AUC, ACC, SENS, SPEC, PPV, NPV and model parameter amount (PARAM). The detailed indicators listed in Table 1 below.

Table 1. The performance comparison of different models in validation cohorts

	AUC	ACC	SENS	SPEC	PPV	NPV	PARAM (Mb)
M1	0.915	0.82	0.8	0.833	0.762	0.862	98
M2	0.862	0.814	0.76	0.853	0.792	0.829	98
M3	0.858	0.797	0.78	0.809	0.75	0.833	172
M4	0.821	0.754	0.76	0.75	0.691	0.81	253
M5	0.768	0.737	0.68	0.824	0.739	0.778	636

M1: model used in manuscript. M2: Resnet50_concatenate. M3: Resnet101. M4: Inception v3.

M5: vgg19.

We draw a radar chart to express more intuitive performance differences, as shown in Fig 1.

Fig 1. The radar chart for comparing different model performance

The model used in manuscript is better in terms of performance and memory usage. Please see page 11, line 235-244.

Q 6: pg 6, line 106- can you comment on this limitation? What fraction of patients would be excluded in general?

A: Breast masses larger than 3.5 cm in diameter were excluded since the width of the probe is approximately 4cm and surrounding normal tissue should be included in the image section to clearly visualize the margin of the mass. Moreover, breast masses deeper than 3 cm in depth were also

excluded due to attenuation of SWE. This information had been added accordingly. Please page 6, line 110-112.

In our center, the fraction of patients whose breast cancers larger than 3.5 cm in diameter or deeper than 3 cm in depth was near 2%.

Q 7: pg 6, line 114 - what was the inter-rater agreement?

A: The result of axillary US evaluated by the experienced radiologists was regarded as positive as long as at least one of the suspicious US findings of ALN was found. Most suspicious US features are quantitative and inter-rater agreement was excellent. To test the intra-observer agreement, one radiologist repeated evaluating the same 30 ALNs at a time interval of 1 week. Inter-observer agreement was investigated by two radiologists, evaluating the same ALNs independently on the same day in another 30 ALNs [1]. Agreement was defined as consistency in positive or negative nodes according to suspicious US features.

The Kappa values were 0.933 for inter-observer agreement and 1 for intra-observer agreement (both $P < 0.001$). This part had been added in the Methods and Results section. Please page 7, line 134-137 and page 12, line 255-256.

Q 8: Did you pre-process your images (besides cropping)? Normalize? Please provide full details of the preprocessing step.

A: We explained the details of image preprocessing in the DLR model part. Before fed into the network, the ROIs were resized to 224*224 pixels and normalized (subtracting the mean of the image by each pixel and divide by the standard deviation). Please see page 7-8, line 158-180.

Q 9: pg 7, line 154. the last 1000 nodes fully connected layer was replaced with our delicate designed three fully connected layers with Xavier initialized weights" - what does delicate design refer to?

A: The specific design of fully connected layer is mainly reflected in two points. Firstly, the number of neurons in the FC layer is optimal for several experiments. i.e. for the first FC layer, we compared the model performance when neurons were 256 or 128 respectively and the results turn out to be better when using 128. Secondly, we embedded two drop out layers with 0.5 ratio between the three FC layers. We have changed 'delicate design' to a more precise expression 'specifically designed'. Please see page 8, line 162-164.

Could you also comment on the reason for the choice of this network architecture? Did you try others? Please provide sufficient technical details regarding the training process.

A: There are two reasons why we choose the ResNet as our base model. Firstly, in many computer vision tasks [2-3], ResNet had proven to be a feature extraction structure with good stability and performance, including medical image classification [4]. Secondly, we compared the model performance when ResNet50, ResNet101, Inception v3 and vgg19 were used as the base model respectively and proved that ResNet50 was the best. The detailed experimental data listed in Table 1. Please see page 11, line 235-244 and page 23, line 585-588.

Can you further clarify the training of the network? Specifically, what did the label consist of? The ground truth for the classification (i.e. one of 3 classes)? What were the range of sized for the ROIs?

A: Before training the model, we preprocessed the image data (ROIs with size 224*224 pixels) and encoded the corresponding pathology type (refers to the classification standard) to one-hot, which was the label. The outputs of the network were used as the classification result, and the cross-entropy of the output and the label was calculated as the loss function. We set learning rate to 1e-4 and applied the Adam optimizer (an improved mini-batch stochastic gradient descent algorithm) to update the model parameters with batch size 32. The maximum iteration step was set to 5000, and the learning rate decayed by 1/2 at 2000 and 4000 steps. Please see page 8, line 176-180.

Q 10: pg 8, line 182 -this section is not clear.

A: We had rewritten this section. Please see page 9, line 195-208.

Q 11: Please comment on the required performance metrics that would make the sentinel node biopsy unnecessary.

A: Although sentinel node biopsy is the standard procedure for the surgery of breast cancer, it is not risk-free. Sentinel node biopsy has false negative rate ranging from 7.8-27.3% [5-7]. In the current study, similar false negative rate was found for clinical parameter constrained DLR to predict axillary lymph node metastases of breast cancers, which indicates clinical parameter constrained DLR has the potential to serve as a noninvasively imaging biomarker to replace sentinel node biopsy for breast cancer, but prospective multicenter validation and long-term follow-up is needed. This had been discussed in the discussion section. Please see page 14, line 311-313, line 322-325.

There were numerous linguistic issues:

Q11: pg 3, line 47 -please define SLND

A: SLND had been defined. Please see page 3, line 43.

There are a number of run-on sentences that could be split up for improved clarity or text that could be rephrased to improve readability including:

1. pg 1, line 14-16

A : The sentence had been changed accordingly. Please see page 1, line 13-16.

2. pg 3, line 59

A : The sentence had been changed accordingly. Please see page 3, line 60-62.

3. pg 3 line 63-65

A : The sentence had been changed accordingly. Please see page 3, line 64-67.

4. pg 4, line 68 This technique showed promising... -> ...showed promise...

A : This sentence had been changed accordingly. Please see page 4, line 70.

5. pg 4, line 69

A : The sentence had been changed accordingly. Please see page 4, line 71-73.

6. pg 4, line 75-77. not sure what "quantificationally" means

A: This sentence means radiomics is an emerging method that can automatically provide a large number of quantitative image features from medical images. The sentence had been changed accordingly. Please see page 4, line 77-78.

7. pg 4, line 90 "... restrain the features extracted from images to make the model more robust." Not sure what "restrain" refers to in this context or how this process makes the model more robust.

A: When there is no clinical information, the classification model has larger decision boundaries, which will increase the risk of model instability. When adding clinical information, the decision boundary of the classification model is limited to a more reasonable range, thus increasing the robustness of the model. We had rewritten this sentence in the text and made a clearer explanation. Please see page 4, line 89-92.

8. pg 5- not sure why this is blank?

A: The format went wrong when the 'word' file converted into the 'pdf' file. This blank page had been deleted.

9. pg7, line 138 ...thought as unqualified... ->rephrase

A: The sentence had been deleted because unqualified 2D-SWE measurements was explained in page 6, line 112-114.

10. pg7, line 156 rephrase "enrolled"

A: We had rewritten this section and rephrased the word. Please see page 8, line 165.

11. pg 8, line 183 - what is "modal" in this context?

A: We had rewritten this section and the 'modal' originally referred to grayscale US images. Please see page 9, line 195-208.

12. pg 8, line 187 -"fc" - please spell out fully connected the first time you use it.

A: FC had been spelt out when presented first time in the manuscript. Please see page 8, line 163.

13. pg 9, line 198 -By using different cut-offs of positive ALNs as reference -> please expand what cut-offs refer to?

A: We had rewritten this section. The cut-off of positive ALNs had been mentioned below, including disease-free axilla (N0) and any axillary metastasis (N+(≥1)), low metastatic burden of axillary disease (N+(1-2)) and heavy metastatic burden of axillary disease (N+(≥3)). Please see page 9, line 211-214.

14. pg 11, line 243

A: The sentence was redundant and was deleted. Please see page 12, line 272.

Reviewer #2:

Introduction

Q1: Line 48, SLN is not controversial, it is standard care at present. Time and expense when combined with resection of the primary tumour is modest.

A: We appreciate the reviewer's comment and the sentence had been revised accordingly. Please see page 3, line 47-51.

Q2: Line 50 "Injury" should be replaced with "complication" and the frequency of the side effects should be stated

A: This had been revised accordingly and the frequency of the side effects had been added. Please see page 3, line 48-51.

Q3: Line 52 replace "can" with "could" and is with was

A: This has been revised accordingly. Please see page 3, line 53.

Q4: Line 64 be more specific. It is only vi status in the list you give which is unknown pre-op

A: We had rewritten this sentence to make it more specific. Please see page 3, line 60-62.

Q5: Line 77 quantificationally is a clumsy word, please rephrase

A: The sentence had been rephrased. Please see page 4, line 77-78.

Q6: Line 78 poor English suggest “when analyzing oncology CT and MRI images” replace “upon” with “on” x2

A: “On” had replaced “upon”. Please see page 4, line 80.

Q7: Line 81 replace “have” with “has”

A: “Has” had replaced “have”. Please see page 4, line 82.

Methods

Q8: 108 please explain what “unqualified SWE images” means

A: “Unqualified SWE images” had been explained in the Methods section. Unqualified 2D-SWE measurements mean little or no shear wave signal is acquired in the ROI of SWE. Please see page 6, line 112-114.

Q9: Were these consecutive lesions? If not, exactly who was recruited and why?

It is not believable that no patient had multiple lesions, in my experience 10-20% of breast cancer patients have multifocal disease on US and 2% have bilateral disease, please explain, if multifocal disease was excluded please state, if the largest lesion only was included please state. If so this needs discussion.

A: These were consecutive breast lesions. Patients with multifocal breast lesions and bilateral disease were excluded because it is difficult to determine which lesion would lead to ALN metastases and which lesion should be chosen to input in the model. Further study was needed to build other model to predict ALN metastases for patients with multifocal breast lesions and bilateral disease. This had been stated in the

exclusion criteria and discussed in the limitation. Please see page 6, line 110; and page 16, line 377-383.

Q10: Were men excluded?

A: Men were excluded.

Q11: Were they symptomatic or mammography detected?

A: About 20% of enrolled patients were symptomatic, mostly for palpating the breast masses. About 10% of patients were detected by mammography, and the remaining patients were detected by ultrasound.

Q12: How many SWE images were performed in how many planes, cancer show marked anisotropy so was any effort made to use the stiffest or softest plane? Multiple images have been shown to increase the reproducibility of the technique.

A: Three SWE images were obtained at the maximal-diameter plane of the breast lesion. Although multiple images have been shown to increase the reproducibility of the technique, only one image would be included. The image with the best quality and fewest artifacts was chosen for analysis. In our future study, multiple SWE images might be included to get better reproducibility. Please see page 7, line 146-147.

Results.

Q13: Please give mean age and age range

A: Mean age and age range were given accordingly. Please see page 11, line 229.

Q14: Line 227 remove “poor” results are data not opinion

A: Poor had been removed. Please see page 11, line 254.

Q15: Clinical data should include only data available pre-operatively. IE core grade not final histological grade, and exclude VI status, US size not histological size.

A: Our clinicopathological data only included data available pre-operatively including age, US size, BI-RADS category, tumor type, ER, PR, HER-2 and Ki-67. We did not include tumor grade because tumor grade shows low concordant rates ranging from 67%-75% between core needle biopsy and surgical excision [9-10]. Please see page 15, line 344-349.

Q16: In patients with a pre-operative diagnosis of axillary metastases (fna or core) how good was prediction of nodal burden at surgery? Especially those who had ANC.? This is a crucial clinical question.

A: In patients with a pre-operative diagnosis of axillary metastases, prediction of nodal burden might mean predicting whether patients had low metastatic burden of axillary disease (N+(1-2)) or heavy metastatic burden of axillary disease (N+(≥3)). This clinical parameter constrained DLR showed excellent prediction ability of nodal burden with an AUC of 0.984 in the training cohort and an AUC of 0.915 in the validation cohort when predicting ALN metastasis between low metastatic burden of

axillary disease (N+(1-2)) and heavy metastatic burden of axillary disease (N+(≥3)). Patients with heavy metastatic burden of axillary disease (N+(≥3)) could undergo ALN dissection directly while patients with low metastatic burden of axillary disease (N+(1-2)) could only undergo SLND. Please see page 12, line 270-281.

Patients who had neoadjuvant chemotherapy were excluded, so the current model could not be used to predict nodal burden for patients who had received NAC.

Discussion

Q17: The Z0011 trial had a number of well publicised limitations which should be mentioned

A: Based on the Z0011 trial, all patients should undergo SLND whether patients have clinically positive node or not. SLN biopsy has some significant limitations, including increasing considerable anesthesia time and expense, and causing complications such as arm numbness or upper limb edema in 3.5%-10.9% of patients [7].

43%–65% of patients who had positive SLNs underwent unnecessary axillary surgery because of no additional non-SLN metastasis, resulting in high morbidity. In addition, patients with negative SLNs have false negative rates ranging from 7.8-27.3% [5-7]. This had been addressed in the Discussion section. Please see page 14, line 309-313.

Q18: A few recent papers have addressed predicting nodal burden with US, these should be mentioned

A: This had been mentioned in the Discussion section. Several studies demonstrated that the number and morphology of abnormal lymph nodes detected by axillary US were predictors of nodal burden, but the diagnostic performances were poor with an AUC of 0.725-0.747 [11,12]. Please see page 14, line 332-335.

Q19: Line 317 tumour grade is routinely estimated from core biopsy samples pre-operatively in most centres, this should be addressed.

A: We had addressed this accordingly and rewritten the sentence. Although tumor grade could be estimated from core biopsy samples pre-operatively, low concordant rates (ranging from 67%-75%) were found between core needle biopsy and surgical excision in previous studies [9-10]. Unlike tumor grade, some pathological data like ER, PR, HER-2 showed high concordant rates of 89%-100% between CNB and surgical specimens [13]. Therefore, in the current study, tumor grade was not included as preoperative pathological data. Please see page 15, line 347-349.

Q20: How would DLR be used in a clinical setting, would an online tool be feasible? Would DLR be possible on uploaded images? How long would it take?

A: We would provide a software about this clinical parameter constrained DLR. This could be an online tool and would be possible on uploaded images. When uploading a file and importing clinical data, we would get the prediction of ALN metastasis. After the model was loaded, the prediction process would take less than 1 second.

Table 1

Q21: HER positivity is usually seen in 10-20% of patients and ER positivity in about 80%, your results especially for HER-2 status of <1% are not credible

A: To improve readability, we had redesigned the Table 1 (Table 2 in the revised paper). In our study, HER-2 positivity takes up 23.2% of patients and ER positivity takes up 80.6% of patients, which was concordant with the epidemiology of breast cancer [13]. In the former Table 1, the number of HER-2 was the weighted average of HER-2 positivity (assigned the value 1) and HER-2 negativity (assigned the value 0). The table 1 had been revised accordingly.

Q22: The percentages in the BIRADS section have been wrongly calculated, each column not line should add up to 100%

A: The percentages in the BIRADS section had been calculated each column. Please see Table 2.

Q23: The location section should be removed

A: The location section had been removed. Please see Table 2.

Q24: The tumor type (ie ductal , lobular etc) should be added

A: The tumor type had been added, included invasive ductal carcinoma, invasive lobular carcinoma and other tumor type. Please see Table 2.

Reference:

- [1] Dan-Dan Li, et al. Combination of two-dimensional shear wave elastography with ultrasound breast imaging reporting and data system in the diagnosis of breast lesions: a new method to increase the diagnostic performance. *European Radiology* 26.9(2016):3290-3300.
- [2] Kaiming H, et al. Mask R-CNN[J]. *IEEE Transactions on Pattern Analysis and Machine Intelligence*, 2018:1-1.
- [3] Szegedy C, et al. Inception-v4, Inception-ResNet and the Impact of Residual Connections on Learning[J]. 2016.
- [4] Xie Y, et al. Knowledge-based Collaborative Deep Learning for Benign-Malignant Lung Nodule Classification on Chest CT[J]. *IEEE Transactions on Medical Imaging*, 2018:1-1.
- [5] Krag D, et al. The Sentinel Node in Breast Cancer — A Multicenter Validation Study[J]. *New England Journal of Medicine*, 1998, 339(14):941-946.
- [6] Krag D N, et al. Technical outcomes of sentinel-lymph-node resection and conventional axillary-lymph-node dissection in patients with clinically node-negative breast cancer: results from the NSABP B-32 randomised phase III trial.[J]. *Lancet Oncology*, 2007, 8(10):881-888.
- [7] Pesek S, et al. The False-Negative Rate of Sentinel Node Biopsy in Patients with Breast Cancer: A Meta-Analysis[J]. *World Journal of Surgery*, 2012, 36(9):2239-2251.
- [8] Sharifi S, et al. Assessment of pathologic prognostic factors in breast core needle biopsies. *Mod Pathol*.1999 Oct ;12(10):941-5.

[9] Harris GC, et al. Correlation of histologic prognostic factors in core biopsies and therapeutic excisions of invasive breast carcinoma. *Am J Surg Pathol*. 2003 Jan ;27(1):11-5

[10] Caudle A S, et al. Predicting the Extent of Nodal Disease in Early-Stage Breast Cancer[J]. *Annals of Surgical Oncology*, 2014, 21(11):3440-3447.

[11] Lim GH, Upadhyaya VS, Acosta HA, Lim JMA, Allen JC, Leong LCH. Preoperative predictors of high and low axillary nodal burden in z0011 eligible breast cancer patients with a positive lymph node needle biopsy result. *Eur J Surg Oncol* 2018;44:945-950

[12] Wang X, Chen L, Sun Y, Zhang B. Evaluation of axillary lymph node metastasis burden by preoperative ultrasound in early-stage breast cancer with needle biopsy-proven metastasis. *Clin Transl Oncol* 2019

[13] Cheang MCU, et al. Ki67 Index, HER2 Status, and Prognosis of Patients With Luminal B Breast Cancer[J]. *JNCI Journal of the National Cancer Institute*, 2009, 101(10):736-750.

Reviewers' Comments:

Reviewer #1:

Remarks to the Author:

Thank you for addressing some of the issues raised in the review. There are still a number of linguistic issues. Additionally, the phrase "clinically regularized DLR" occurs throughout including in the graphs.

Please provide the architecture of your full model including the 3 FC layers.

It is still not clear how you went from a segmentation mask/ROI to a label used for the network. If any part of the ROI had a positive label, was the whole patch positive?

Did you have a training, validation and test sets or just training and validation? How did you decide stopping criteria?

Given the drop in performance on the validation set for the DLR on images only, it is not clear that there training was stopped at an appropriate place - seems like overfitting?

Reviewer #2:

Remarks to the Author:

The authors have responded thoughtfully and to the questions raised by the referees.

November 27, 2019

RE: NCOMMS-19-24501B

TITLE: Deep learning radiomics of conventional ultrasound and shear wave elastography in early-stage breast cancer: preoperative prediction of axillary lymph node status.

Dear reviewers,

Thank you for reviewing our manuscript entitled “Deep learning radiomics of conventional ultrasound and shear wave elastography in early-stage breast cancer: preoperative prediction of axillary lymph node status”. As per your request, this letter will address all the raised criticisms. Modifications were made and highlighted in the revised text in response. Reviewers' comments and answers:

Reviewer #1:

Q1: There are still a number of linguistic issues. Additionally, the phrase "clinically regularized DLR" occurs throughout including in the graphs.

A: Thank you for your kindly reminding. We have modified the linguistic issues and have checked the manuscript carefully. All modifications were highlighted in the revised text.

Q2: Please provide the architecture of your full model including the 3 FC layers.

A: The detailed structure of the model is shown in table S1. And this table has also been added to the supplementary materials. Please see page 8 , line 164-165.

Table S1 detailed architecture of the network

Layer name	Output size	DLR
Conv1	112×112	$7 \times 7, 64, \text{stride } 2$
Conv2_x	56×56	$\begin{bmatrix} 1 \times 1, 64 \\ 3 \times 3, 64 \\ 1 \times 1, 256 \end{bmatrix} \times 3$
Conv3_x	28×28	$\begin{bmatrix} 1 \times 1, 128 \\ 3 \times 3, 128 \\ 1 \times 1, 512 \end{bmatrix} \times 4$
Conv4_x	14×14	$\begin{bmatrix} 1 \times 1, 256 \\ 3 \times 3, 256 \\ 1 \times 1, 1024 \end{bmatrix} \times 6$
Conv5_x	7×7	$\begin{bmatrix} 1 \times 1, 512 \\ 3 \times 3, 512 \\ 1 \times 1, 2048 \end{bmatrix} \times 3$
Fc6	1×1	Average pool, 256 – d fc
Fc7	1×1	64 – d fc
Fc8	1×1	2 – d fc, softmax

*Note: t the gray shading (Fc7) refers to deep feature output layer.

Q3: It is still not clear how you went from a segmentation mask/ROI to a label used for the network. If any part of the ROI had a positive label, was the whole patch positive?

A: Thank you for reminding us of this ambiguity. Rectangular ROIs containing the entire tumor regions were defined by five radiologists with 18, 5, 3, 2, and 2 years of experience, respectively. The term ‘ROI patches’ in the original manuscript may lead to some misunderstandings. We have changed ‘ROI patches’ into ‘rectangular ROIs’ in the revised manuscript. In this way, the labels of images and ROIs are the same. Please see page 8, line 167 and 170.

Q4: Did you have a training, validation and test sets or just training and validation? How did you decide stopping criteria?

A: We divided the enrolled patients into training and validation sets in the ratio of 4:1 in the original manuscript. The stopping criteria was referred to following radiomics literatures (Ref.1 – Ref.5). In a large number of clinical studies, the data set is divided into discovery cohort (the training set in this paper) and validation cohort (the validation set in this paper). In the training stage, the training loss and validation loss trends to decrease. The training process was considered complete when the validation loss trends to be stable (three consecutive recording losses are not decrease). For example, the loss over time of prediction of ALN metastasis (N0 v.s. N+) is shown in Fig.1.

Figure S1. Training losses and validation losses of CNN model

Thanks to the reviewer for raising this question. In order to further verify the stability of the model, we added independent testing

experiments to the revised manuscript. We randomly choose 1/4 training samples to form a cross-validation set and use the original validation set as an independent testing set in the tasks of prediction of ALN metastasis (N0 v.s. N+ and N+(1-2) v.s. N+(≥3)). The final ratio of training: validation: testing is 3:1:1. We use the above training strategy to train the model, and the experimental results are shown in table S2 and table S3. Experimental results show that this training strategy is also effective for independent testing. The independent testing results were supplemented into the supplementary materials. Please see page 12, line 267-268 and 281-282.

Table S2. performance of clinical parameter constrained DLR model in predicting ALN metastasis (N0 v.s. N+(≥1))

	AUC	ACC	SENS	SPEC	PPV	NPV
Training	0.929	0.841	0.807	0.866	0.815	0.860
Validation	0.894	0.822	0.760	0.868	0.809	0.831
Testing	0.874	0.797	0.700	0.868	0.796	0.797

Note: AUC=area under the receiver operating characteristic curve; ACC=accuracy; SENS=sensitivity; SPEC=specificity; PPV= positive predict value; NPV= negative predict value; DLR= deep learning radiomics; Training = training cohort; Validation = validation cohort; Testing =testing cohort

Table S3. performance of clinical parameter constrained DLR model in predicting ALN metastasis (N+(1-2) v.s. N+(≥3))

	AUC	ACC	SENS	SPEC	PPV	NPV
Training	0.966	0.884	0.947	0.844	0.794	0.962
Validation	0.908	0.880	0.950	0.833	0.792	0.962
Testing	0.895	0.837	0.842	0.833	0.762	0.893

Note: AUC=area under the receiver operating characteristic curve; ACC=accuracy; SENS=sensitivity; SPEC=specificity; PPV= positive predict value; NPV= negative predict

value; DLR= deep learning radiomics; Training = training cohort; Validation = validation cohort; Testing =testing cohort

Q5: Given the drop in performance on the validation set for the DLR on images only, it is not clear that there training was stopped at an appropriate place - seems like overfitting?

A: The training stopping criteria was explained in Q4. In the task of prediction of ALN metastasis, it is difficult to distinguish the number of lymph node metastasis ($N+(1-2)$ v.s. $N+(\geq 3)$) because all the images are closing to the decision boundary (all the samples have lymph node metastasis). Therefore, the small inter-class variance makes it easy to be overfitting if we only use the image features. Clinical information itself has a certain ability of classification, adding to deep learning can increase the inter-class variance. On the other hand, the constraints of clinical information make the search space of network parameters reduce, greatly reducing the risk of over fitting. Therefore, the model combining image information and clinical information is better than the model only relying on image, which is also the innovation of our method.

Reference:

Ref [1] Wang, K., et al., Deep learning Radiomics of shear wave elastography significantly improved diagnostic performance for assessing liver fibrosis in chronic hepatitis B: a prospective multicentre study. Gut, 2019. **68**(4): p. 729-741.

Ref [2] Lao, J.W., et al., A Deep Learning-Based Radiomics Model for Prediction of Survival in Glioblastoma Multiforme. Scientific Reports, 2017. **7**.

- Ref [3] Badgeley, M.A., et al., Deep learning predicts hip fracture using confounding patient and healthcare variables. *Npj Digital Medicine*, 2019. **2**.
- Ref [4] Mullooly, M., et al., Application of convolutional neural networks to breast biopsies to delineate tissue correlates of mammographic breast density. *NPJ Breast Cancer*, 2019. **5**: p. 43.
- Ref [5] Dominietto, M., et al., Deep Learning Based on Radiomics Features Dataset to Predict the Outcome of Skull-Base Chordomas Patients Treated with Pencil Beam Scanning Proton Therapy. *International Journal of Radiation Oncology Biology Physics*, 2019. **105**(1): p. E134-E134.

Reviewers' Comments:

Reviewer #1:

Remarks to the Author:

Thanks for addressing the points raised by the reviewers.

The phrase "Clinical parameter constrained DLR " is still somewhat unusual. It is still not clear how you are "constraining" the DL model by the clinical parameters.

Based on the methods section of the abstract, it is not clear what images were used for the deep learning (conventional, axillary or SWE) - please clarify.

Not clear what you mean by

Breast images were analyzed by DLR and classified by clinical information ... -> can you clarify what you mean by "classified by clinical information"?

Please provide of the results for the test set in the abstract - that is much more important than the training or validation sets. Please move the results of the test set to the main part, not just the supplement. Table 3 should have the test set results.

Did you compare the performance of just the US or the SWE + clinical parameters (not just the combined architecture)?

Did you try appending the clinical parameters to the last layer of the neural network (instead of using an SVM)?

Since the "SWE was performed thrice" - did you study the impact of the choice of the image?

Instead of two parallel networks for the US and SWE, did you try a 2 channel approach? The parallel nature of the network is not clear in figure 2.

Can you confirm that the image only results were with the parallel architecture?

Reporting the results of the training dataset (instead of a completely independent dataset leads to overly optimistic assessment of the performance. The training and validation can be moved to the supplement if necessary but it is imperative that the test performance is reported in the main manuscript.

The results of the visualization are very qualitative. Can you quantify them?

Since you performed boot-strap analysis, please provide those confidence intervals, not just point estimates for the results.

Please clarify that in order to use the algorithm, a radiologist would have to first draw an ROI around the node.

We hypothesis that clinical parameter constrained DLR... - > We **hypothesize** that... or **Our hypothesis** is ...

Contrasting with hand craft and engineered features ... -> .. **hand-crafted**...

Jan 16, 2020

RE: NCOMMS-19-24501C

TITLE: Deep learning radiomics can predict axillary lymph node status in early-stage breast cancer

Dear reviewers,

Thank you for reviewing our manuscript entitled "Deep learning radiomics can predict axillary lymph node status in early-stage breast cancer". As per your request, this letter will address all the raised criticisms. Modifications were made and highlighted in the revised text in response. Reviewers' comments and answers:

Reviewer #1:

Q1: The phrase "Clinical parameter constrained DLR " is still somewhat unusual. It is still not clear how you are "constraining" the DL model by the clinical parameters.

A: Thank you for your consideration and question. To avoid misunderstanding, we changed the "clinical parameter constrained DLR" to "clinical parameter combined DLR". In order to clarify the pipeline of our method as clear as possible, we redraw the method schematic diagram, and made a more detailed explanation in the revised paper. Please see Figure 6 and its figure legend. As shown in following Figure A, in the training stage, the deep learning model receives images as inputs, and updates the parameters using the image information. In the inference stage, the trained model encodes images as deep-features, which combined with clinical parameters to make the final decision. By changing the feature distribution on high dimensional manifold, the clinical parameters constrain the results of the DL model.

Figure A. The overall pipeline of the model. The parallel pre-trained ResNet model encodes the input images to features which combined with clinical parameters and further be classified by an SVM model. The detailed parameters of the network are summarized in Supplemental Table 1 in the Supplementary Information file.

Q2: Based on the methods section of the abstract, it is not clear what images were used for the deep learning (conventional, axillary or SWE) - please clarify.

A: Conventional US images and SWE images of breast masses were used for deep learning. The method section of the abstract has been rewritten. Please see Page 2, Line 30-33.

Q3: Not clear what you mean by Breast images were analyzed by DLR and classified by clinical information ... -> can you clarify what you mean by "classified by clinical information"?

A: Thanks for pointing this out. This sentence has been deleted to avoid misunderstanding. Clinical information includes clinicopathologic data and axillary US findings which is described in detail in the Data Analysis section of

the manuscript. Please see Page 14, Line 333-339. The clinical information is combined with the deep features to collaboratively train an SVM classifier for predicting ALN status.

Q4: Please provide of the results for the test set in the abstract - that is much more important than the training or validation sets. Please move the results of the test set to the main part, not just the supplement. Table 3 should have the test set results.

A: Thank you for your kind reminder and we have changed the abstract and Table 3 according to your suggestion.

Q5: Did you compare the performance of just the US or the SWE + clinical parameters (not just the combined architecture)?

A: Thank you for your advice. We added some experiments to compare the performance of just using the US or the SWE + clinical parameters. The results are summarized in following Table A. This has been added in Supplementary Note 1 in the Supplementary Information file and explain it on the Page 7, Line 156-160 of the revised manuscript. The experimental results show poor prediction of ALN metastasis (N0 and N+(≥1)) using single modal image + clinical parameters.

Table A. the results of using single modal image and clinical parameters to predict ALN metastasis (N0 and N+(≥1))

Methods		AUC	ACC (%)	SENS (%)	SPEC (%)	PPV (%)	NPV (%)
Experiment1	T	0.800	72.2	71.6	73.8	72.7	72.7
		[0.757, 0.843]	[67.3, 76.7]	[64.7, 77.8]	[67.1, 79.7]	[65.8, 78.8]	[66.0, 78.7]
	V	0.767	71.6	67.4	73.1	64.7	75.4
		[0.675, 0.858]	[63.3, 79.8]	[52.5, 80.1]	[60.9, 83.2]	[50.1, 77.6]	[63.1, 85.2]
	I-T	0.752*	72.0	63.3	79.4	68.9	75.0
		[0.657, 0.847]	[63.0, 79.4]	[48.3, 76.6]	[67.9, 88.3]	[53.4, 81.8]	[63.3, 84.5]
Experiment2	T	0.790	72.2	73.1	71.8	71.6	73.2
		[0.745, 0.834]	[67.3, 76.7]	[66.3, 79.2]	[65.0, 77.9]	[64.9, 77.8]	[66.5, 79.3]
	V	0.750	69.8	71.4	68.7	62.5	76.7
		[0.653, 0.846]	[60.5, 77.4]	[56.7, 83.4]	[56.2, 79.4]	[48.5, 75.1]	[63.8, 86.7]

I-T	0.731**	68.6	70.0	63.2	58.3	74.1
	[0.636, 0.826]	[59.4, 76.2]	[55.4, 82.1]	[50.7, 74.6]	[44.9, 70.9]	[61.0, 84.7]

Note: Experiment1 = US + clinical parameters; Experiment2 = SWE + clinical parameters. 95% confidence intervals are included in brackets. AUC=area under the receiver operating characteristic curve; ACC=accuracy; SENS=sensitivity; SPEC=specificity; PPV= positive predict value; NPV= negative predict value; PARAM= model parameters amount; T = training cohort (n=350); V = validation cohort (n=116), I-T = independent test cohort (n=118).

* indicates $P=0.006$, Delong et al. in comparison with clinical parameter combined DLR in independent test cohort.

** indicates $P=0.002$, Delong et al. in comparison with clinical parameter combined DLR in independent test cohort.

Q6: Did you try appending the clinical parameters to the last layer of the neural network (instead of using an SVM)?

A: We have tried appending the clinical parameters to the last layer of the neural network. Please see Table 1 in the revised manuscript. The results appending the clinical parameters to the last layer of the Resnet50 are summarized in the line of Resnet50 concatenate.

Q7: Since the "SWE was performed thrice" - did you study the impact of the choice of the image?

A: This Siemens S2000 ultrasound scanner (Siemens Healthineers, Mountain View, CA, USA) provides quality map to evaluate the quality of SWE. The quality map was displayed in red-yellow-green to represent low-intermediate-high quality respectively. The quality map was obtained to evaluate the quality of the SWE first. Then the velocity map of SWE was obtained. Guided by the quality map, the SWE image with fewest artifacts and the best quality was stored and chosen for analysis. This has been stated in the revised manuscript. Please see Page 14, Line 326-331.

Because the optimal SWE data for single measurement was stored and used for

analysis, we were not able to compare the effect of image selection on the results in this study. Thank you for your suggestion. This is a very interesting question. In future research, we will pay attention to storing multiple measurement images to study the influence of image selection on model results.

Q8: Instead of two parallel networks for the US and SWE, did you try a 2 channel approach? The parallel nature of the network is not clear in figure 2.

A: Thank you for your suggestion. Two channel approach is indeed a good idea, but this cannot be implemented in our experiments. Because we are using a transfer learning strategy, the number of input image channels must match the requirements of the pre-trained model. In fact, the structure of two parallel networks has proven to be an efficient and practical method for multi-modal or multi-view fusion analysis^{1,2}.

Q9: Can you confirm that the image only results were with the parallel architecture?

A: In the deep learning radiomics modeling, parallel architecture was used when the inputs were B-mode and SWE images. By following your suggestion, we added the experiments of one modality image combined with clinical information in this edition. When one modality was used as the input, one path of parallel architecture was used to obtain the deep-feature of the image.

Q10: Reporting the results of the training dataset (instead of a completely independent dataset leads to overly optimistic assessment of the performance. The training and validation can be moved to the supplement if necessary but it is imperative that the test performance is reported in the main manuscript.

A: Thank you for your advice. We have changed tables of results according to your suggestion. Please see Table 1, Table 3, Table 4 in the revised manuscript.

Q11: The results of the visualization are very qualitative. Can you quantify them?

A: The previous Figure 6 (now Figure 5 in the revised manuscript) has been changed accordingly. We add a color bar ranging from 0 to 1 for pseudo colors for quantitative analysis.

Q12: Since you performed boot-strap analysis, please provide those confidence intervals, not just point estimates for the results.

A: Thank you for your advice. The 95% confidence intervals have been added in the revised manuscript. Please see Table 1, Table 3, Table 4 in the revised manuscript.

Q13: Please clarify that in order to use the algorithm, a radiologist would have to first draw an ROI around the node.

A: Thanks for your suggestion. We have implemented this description in the revised manuscript. Please see Page 16 Line 392-393.

Q14: We hypothesis that clinical parameter constrained DLR... -> We *hypothesize* that... or *Our hypothesis* is ...

A: This sentence has been changed accordingly. Please see page 4, line 95. Thank you.

Q15: Contrasting with hand craft and engineered features ... -> .. *hand-crafted*...

A: This sentence has been changed accordingly. Please see page 15, line 368. Thank you.

Reference:

[1] NIE D, LU J F, ZHANG H, et al. Multi-Channel 3D Deep Feature Learning for Survival Time Prediction of Brain Tumor Patients Using Multi-Modal Neuroimages [J]. Sci Rep-Uk, 2019, 9

[2] CHEN X Z, MA H M, WAN J, et al. Multi-View 3D Object Detection Network for Autonomous Driving [J]. 30th IEEE Conference on Computer Vision and Pattern Recognition (Cvpr 2017), 2017, 6526-34.